# Evaluation of the diagnostic accuracy of two point-of-care tests for COVID-19 when used in symptomatic patients in community settings in the UK primary care COVID diagnostic accuracy platform trial (RAPTOR-C19)

Brian D. Nicholson[1]*, Philip J. Turner[1], Thomas R. Fanshawe[1], Alice J. Williams[1], Gayatri Amirthalingam[2], Sharon Tonner[1], Maria Zambon[3,4], Richard Body[5,6,7], Kerrie Davies[8,9], Rafael Perera[1‡], Simon de Lusignan[1‡], Gail N. Hayward[1‡], F.D. Richard Hobbs[1‡], on behalf of the RAPTOR-C19 Study Group and the CONDOR Steering Committee[¶]

1 Nuffield Department of Primary Care Health Sciences, University of Oxford, Oxford, United Kingdom, 2 Immunisation and Vaccine Preventable Diseases Division and Public Health Programmes, UK Health Security Agency, London, United Kingdom, 3 Influenza and Respiratory Virology & Polio Reference Service, UK Health Security Agency, London, United Kingdom, 4 NIHR Health Protection Research Unit, Imperial College London, London, United Kingdom, 5 Division of Cardiovascular Science, University of Manchester, Manchester, United Kingdom, 6 Emergency Department, Manchester Royal Infirmary, Manchester University NHS Foundation Trust, Manchester, United Kingdom, 7 Manchester Academic Health Science Centre, & Healthcare Sciences Department, Manchester Metropolitan University, Manchester, United Kingdom, 8 Healthcare Associated Infections Research Group, Leeds Teaching Hospitals NHS trust and University of Leeds, Leeds, United Kingdom, 9 NIHR Leeds MedTech In vitro Diagnostic Co-operative, Leeds Teaching Hospitals NHS Trust and University of Leeds, Leeds, United Kingdom

☯ These authors contributed equally to this work.
‡ RP, SL, GNH, and FDRH also contributed equally to this work.
¶ Membership of the RAPTOR-C19 study group and the CONDOR Steering Committee are provided in the Acknowledgments.
* brian.nicholson@phc.ox.ac.uk

## Abstract

### Background and objective

Point-of-care lateral flow device antigen testing has been used extensively to identify individuals with active SARS-CoV-2 infection in the community. This study aimed to evaluate the diagnostic accuracy of two point-of-care tests (POCTs) for SARS-CoV-2 in routine community care.

### Methods

Adults and children with symptoms consistent with suspected current COVID-19 infection were prospectively recruited from 19 UK general practices and two COVID-19 testing centres between October 2020 and October 2021. Participants were tested by trained healthcare workers using at least one of two index POCTs (Roche-branded SD Biosensor Standard™ Q SARS-CoV-2 Rapid Antigen Test and/or BD Veritor™ System for Rapid Detection of SARS-CoV-2). The reference standard was laboratory triplex reverse

**Data Availability Statement:** Data cannot be shared publicly because of participant confidentiality considerations. Research data access requests should be submitted to the Nuffield Department of Primary Care Health Sciences Information Guardian for consideration (contact via information.guardian@phc.ox.ac.uk) for researchers who meet the criteria for access to confidential data.

**Funding:** This study was funded by the following grants: University of Oxford Medical Sciences Division Benefactors Urgent COVID-19 Fund (COVID-19 Research Response Fund Grant 0009325 - https://researchsupport.admin.ox.ac.uk/funding/internal?filter-566-funding%20opportunity%20type-451761=4021&filter-1686-status-451761=9876) to BDN, SdL, FDRH, JJL, TRF, PJT, GNH, GA, MZ, AD, UH The National Institute for Health and Care Research (NIHR) School for Primary Care Research (SPCR grant 495 - https://www.spcr.nihr.ac.uk/career-development/funding) to BDN, SdL, FDRH, GNH, PJT, JJL, TRF, UH Urgent Public Health funding received by the CONDOR platform from the NIHR and Asthma + Lung UK (NIHR UPH grant COV0051 - https://www.nihr.ac.uk/researchers/funding-opportunities/) to BDN, PJT, RB, KD, GNH, RP, PB, DAP, PD, MW, AJA The three sources of funds above are the sources of funds which specifically funded the study. The following disclosure of funds did not specifically fund the study, but part supported some of the staff who took part and thus need to be declared: TRF, LM, GNH, PJT, RP, GE and FDRH received funding from the NIHR Community Healthcare MedTech and In Vitro Diagnostics Co-operative at Oxford Health NHS Foundation Trust (MIC-2016-018). FDRH, RP and TRF received funding from the NIHR Applied Research Collaboration Oxford and Thames Valley at Oxford Health NHS Foundation Trust. RP acknowledges part support from the Oxford Martin School. JJL is funded by the NIHR (Doctoral Research Fellowship NIHR300738). AAM is funded by a Wellcome Trust Doctoral Fellowship. AES is funded by an NIHR Academic Clinical Fellowship (ACF-2019-13-009). AJW acknowledges an Enriching Engagement grant from the Wellcome Trust for PPI work on RSC general surveillance.

**Competing interests:** We have read the journal's policy and the authors of this manuscript have the following competing interests: The authorship declares funding support for this study from the University of Oxford Medical Sciences Division Benefactors Urgent COVID-19 Fund, the National Institute for Health and Care (NIHR) School of Primary Care Research, and Urgent Public Health

transcription quantitative PCR (RT-PCR) using a combined nasal/oropharyngeal swab. Diagnostic accuracy parameters were estimated, with 95% confidence intervals (CIs), overall, in relation to RT-PCR cycle threshold and in pre-specified subgroups.

## Results

Of 663 participants included in the primary analysis, 39.2% (260/663, 95% CI 35.5% to 43.0%) had a positive RT-PCR result. The SD Biosensor POCT had sensitivity 84.0% (178/212, 78.3% to 88.6%) and specificity 98.5% (328/333, 96.5% to 99.5%), and the BD Veritor POCT had sensitivity 76.5% (127/166, 69.3% to 82.7%) and specificity 98.8% (249/252, 96.6% to 99.8%) compared with RT-PCR. Sensitivity of both devices dropped substantially at cycle thresholds ≥30 and in participants more than 7 days after onset of symptoms.

## Conclusions

Both POCTs assessed exceed the Medicines and Healthcare products Regulatory Agency target product profile's minimum acceptable specificity of 95%. Confidence intervals for both tests include the minimum acceptable sensitivity of 80%. In symptomatic patients, negative results on these two POCTs do not preclude the possibility of infection. Tests should not be expected to reliably detect disease more than a week after symptom onset, when viral load may be reduced.

## Registration

ISRCTN142269.

## Introduction

As point-of-care tests (POCTs), lateral flow device antigen (LFD-Ag) tests provide rapid results that avoid the delays and costs associated with laboratory testing [1] and may be used for community testing for SARS-CoV-2. They provide decentralised, near-real-time information to guide individual decisions about self-isolation and treatment, enabling enhanced surveillance of health and social care staff with potential to reduce community transmission through early detection. For use in primary care, the ideal test would be simple to use with minimal training required, give rapid but accurate results, and present a low biosafety risk. As many countries are reducing or withdrawing community testing at dedicated testing centres, testing for SARS-CoV-2 is falling to community-based healthcare workers, such as those working in General Practice, and to patients.

There are concerns about the diagnostic accuracy of LFD-Ag devices and in particular LFD-Ag test performance when used by front-line community-based healthcare workers in usual care settings. False negatives are more damaging in the community as ambulatory patients can potentially propel community transmission, whilst false positives in otherwise healthy individuals could hamper efforts to maintain employment and education and result in inappropriate management [2].

Whilst the evidence base for LFD-Ag SARS-CoV-2 testing has steadily increased since early 2020 [3], community settings, where most tests take place, are less well studied. Extrapolating results from one clinical setting or population to another risks spectrum bias and is not

funding for the CONDOR Platform from the NIHR and Asthma+Lung UK. The RAPTOR-C19 study team received analysers and assays free of charge from Becton Dickinson for evaluation in this study. GNH declares funding from the National Institute for Health and Care Research (NIHR) paid to the University of Oxford. KD declares grant funding from Alere Inc and Cepheid Inc paid to her institution for unrelated research. TRF declares NIHR support from the NIHR Community Healthcare MIC for diagnostic evaluation research. AJW declares grant funding received by the University of Oxford through a Wellcome Trust Enriching Engagement grant which has supported unrelated patient participation work carried out by the Royal College of General Practitioners Research Surveillance Centre (based at the University of Oxford) for surveillance work. GE declares funding support from the NIHR Community Healthcare MIC received by the University of Oxford. AM declares the support of a Wellcome Trust Doctoral Research Fellowship and an NIHR In-practice Fellowship unrelated to this research. PJT declares support from the NIHR Community Healthcare MIC for diagnostic evaluation research. PJT has provided expert support to the Longitude Prize AMR competition administration which is unrelated to this project and for which the University of Oxford received an honorarium. RB declares grant funding for this project from the NIHR and Asthma+Lung UK, with additional funding from the Department of Health and Social Care paid to his host institution. RB declares grants from Siemens Healthineers, Abbott Point-of-Care and Ancon, all paid to his institution for unrelated research. He declares consulting fees received by his institution from Roche, Siemens, Aptamer Group, LumiraDx, Beckman Coulter and Radiometer, with personal fees received from Psyros Diagnostics. RB has received support for attending meetings / travel from Roche and EMCREG International. RB has participated on data safety monitoring boards or advisory boards for the unrelated FORCE Trial, REWIRE Trial, TARGET-CTA, and Magnetocardiography study (MAGNETIC - sponsored by Creavo). RB is the Deputy National Specialty Lead for Trauma & Emergency Care, National Institute for Health and Care Research Clinical Research Network. RB declares receipt of donated reagents for research not detailed in this paper from Roche, LumiraDx, BD, iXensor, Abbott Point-of-Care, Randox, Avacta, Menarini, loan of analysers from Randox and Menarini, and assays run free of charge for research purposes by Chronomics, My110, and Ancon. JJL declares funding from an NIHR Doctoral Research Fellowship which is unrelated to this research. LM

recommended [4]. In-context evaluations reflect the dynamics of disease transmission, the capabilities of those performing the test and the circumstances under which they are operating. Community populations have a relatively low prevalence and severity of disease, there is overlap in symptomatic presentation with other common clinical syndromes, and the population includes many elderly and frail patients who may mount weaker immune responses to circulating respiratory virus. Studies of selective patient samples tested within laboratories by highly trained staff, or hospital populations who are likely to be more severely unwell, have differing viral loads, and may undergo invasive procedures to increase yield of respiratory tract sampling. Community staff performing POCTs often have little-or-no laboratory experience and no ready access to technical support. Therefore, data on performance of diagnostics in the community is important to inform clinical decisions in the main area of use for these tests.

We aimed to conduct a community based prospective diagnostic accuracy study of POCTs for SARS-CoV-2 infection in symptomatic patients performed by front-line healthcare workers.

## Methods

### Design

RAPTOR-C19 (RApid community Point-of-care Testing fOR COVID-19) is the community testbed for diagnostic testing for SARS-CoV-2 within the UK's COVID-19 National DiagnOstic Research and Evaluation Platform (CONDOR) [5]. It was designed as a prospective platform diagnostic accuracy study, conducted in the community, for the assessment of diagnostic accuracy of point-of-care tests (POCTs) for SARS-CoV-2 infection. RAPTOR-C19 allows for POCTs that test for either active or past infection; the present paper relates to the first two POCTs assessed via this study, both of which test for active infection. Further diagnostic tests are undergoing assessment within the platform. The published protocol gives full details of the study design [6], and a summary is provided here.

### Ethical approval

This study was approved by the North West-Liverpool Central Research Ethics Committee (20/NW/0282). Participants were provided with information about the study via electronic participant information accessible online. All participants (or their parent or guardian, where applicable) gave informed consent via an e-consent process conducted online to minimise the risk of disease transmission, with the completed consent form emailed to the participant.

### Recruitment and participant eligibility

The main setting for this study was UK primary care. Nineteen general practices were recruited after email invitation for expressions of interest, following the sharing of Research Information Sheets for GP surgeries to practices identified through the Oxford-Royal College of General Practitioners (RCGP) Research and Surveillance Centre (RSC) and the National Institute for Health and Care Research (NIHR) Clinical Research Network (CRN). To increase recruitment, two COVID-19 community testing centres for symptomatic individuals were added as additional recruitment sites in the spring of 2021. Participants were adults and children presenting with symptoms of active infection consistent with suspected current COVID-19 (see [6] for a list of specific symptoms). Within these criteria, practices may have differed in their approaches to recruitment (for example, some may not have had capacity to recruit participants on certain days of the week), and so included participants can not necessarily be considered as a consecutive series among those eligible and willing to participate.

## Baseline and follow-up assessments

In addition to undergoing testing for the index test(s) and reference standard described below, further participant information was collected at the time of recruitment using an electronic case report form (eCRF). Variables included age, sex, ethnicity, presence and duration of specified symptoms within the preceding 14 days, vaccine status, household contacts diagnosed with SARS-CoV-2, and the timing and results of previous tests for SARS-CoV-2. Adult participants (age $\geq$ 16 years) recruited from general practices were asked to provide a venous blood sample for antibody testing, collected by appropriately trained staff. These participants were also invited to attend a second visit, or be visited at home by a research nurse, after 28 days to provide a second sample for repeat antibody testing. Adult participants were asked to complete an online daily symptom diary for 28 days after recruitment, but as completion rates were low, no diary data are reported here. Linked electronic health records provided collateral information about subsequent hospitalisation, SARS-CoV-2 test results (in addition to those performed for this study) and mortality within 28 days. Serious adverse events related to test usage were reported by study sites to the RAPTOR-C19 coordination centre via adverse events reporting forms, which were evaluated by clinical staff.

## Index tests

This paper presents results from two POCTs. The SD Biosensor Standard™ Q SARS-CoV-2 Rapid Antigen Test (REF 9901-NCOV-01G, branded and distributed by Roche Diagnostics GmbH, Mannheim, Germany) was used from the start of the study. This test incorporates an internal quality control and is read and interpreted manually by the user following the prescribed assay incubation period [7].The BD Veritor™ System for Rapid Detection of SARS--CoV-2 coupled with the BD Veritor™ Plus Analyzer (REF 256089, Becton Dickinson and Company, Maryland, USA) was used from January 2021 onwards. This assay also incorporates an internal quality control but differs from the SD Biosensor test, as results are read by the associated analyser following either a manually timed incubation process (Analyze Now mode) or in an automated manner (Walk Away mode) which times incubation and reads automatically [8]. Both assays consist of individually packaged LFD cassettes with associated swabbing and sample extraction materials. The RAPTOR-C19 and CONDOR teams deemed both tests could feasibly be used by community healthcare workers, including those without clinical qualifications, with minimal training. Both tests had a buffer with SARS-CoV-2 inactivation capacity and a process with no associated aerosol generating procedure for use away from the laboratory. Recruitment sites used either one or both POCTs, depending on availability. Some participants were tested using both POCTs, and as each candidate POCT used a different sampling site (nasopharyngeal for SD Biosensor, nasal for BD Veritor), order of sampling was judged unlikely to disadvantage either POCT. Index test results were neither shared with the patient nor used as a basis for clinical decision-making. Clinical site staff, including general practitioners, nurses and healthcare assistants, took the samples. They received training via a webinar before recruiting to the study, and were asked to adhere to the manufacturers' instructions for use. Only manufacturer-issued swabs and materials were used to collect and process samples. Index test results were recorded in the eCRF by site staff as 'Positive', 'Negative', or 'Unknown/No result'. Further details of testing procedures are provided as Supplemental Material.

## Reference standard

The reference test for active infection was an in-house validated reverse transcription quantitative PCR (RT-PCR) for the detection of ORF1ab and E gene regions of SARS-CoV-2. The

assay incorporated ORF1ab primers and probes as published by the Chinese Center for Disease Control and Prevention and E gene primers and probe published by Corman et al. [9, 10]. The assay used the ThermoFisher TaqPath 1-Step Multiplex Master Mix (ThermoFisher Scientific, Waltham, Massachusetts, United States) carried out on the ABI QuantStudio 7 flex real-time PCR system (Applied Biosystems Corporation, Waltham, Massachusetts, United States). Testing was performed at the same Public Health England (latterly UK Health Security Agency) laboratory, using a combined nasal/oropharyngeal swab taken during the same visit as when the index test(s) were performed. Results were reported as positive or negative for SARS-CoV-2, with RT-PCR cycle threshold (Ct) values provided for each assay target [11]. The reference test was conducted blind to the results of the index tests. Reference results were not available for at least 24 hours after recruitment. The reference sample was also tested for respiratory syncytial virus (RSV), human metapneumovirus (hMPV), seasonal coronavirus, and influenza. We linked baseline and POCT data to date-matched reference standard data using a unique patient identifier. During the earlier phase of the study, delivery delays and recording and administrative errors meant that for some participants no reference swab was analysed. For some others the swab used for RT-PCR could not be reliably date-matched with the recruitment date of the participant. These participants were excluded from the primary analysis but included in a sensitivity analysis.

## Sample size

The sample size was based on a target of 150 reference standard positive individuals, for each index test. If the true sensitivity of an index test were 90% or higher, a similar number of positive samples (144) would yield a standard error of the estimate of the sensitivity of ≤2.5%, and a 95% confidence interval width of ±5%. This would have ≥90% power to detect a difference from a level of 80% sensitivity (the level specified as "desired" by the MHRA Target Product Profile [12]), at a 5% significance level. Based on an assumed prevalence of 10%, the original target total sample size was 1500 (see Statistical Analysis Plan in Supplemental Materials and published protocol for full details). In the event, the observed prevalence was higher than expected and the study was terminated when the 150 positive sample target was met. For the SD Biosensor POCT, an interim analysis for futility (i.e. to test if sensitivity and specificity estimates fell below pre-specified thresholds) was performed at the end of July 2021 using data from the first 331 participants recruited. As this did not lead to discontinuation for futility, recruitment continued until the full target sample size was exceeded. Details are available as Supplemental Materials.

## Statistical methods

A Statistical Analysis Plan was written before the analysis was performed and is available in Supplemental Material. We calculated the prevalence of positive RT-PCR results, and the sensitivity, specificity and predictive values for each index test alongside exact 95% confidence intervals. Index test results were presented graphically in relation to the RT-PCR cycle threshold. Prespecified subgroup analyses split results by participant characteristics including age, sex, ethnicity, spectrum of disease, recruitment method and recruiting practice. We also performed a post-hoc analysis of diagnostic performance against time since symptom onset. We summarised recruitment rates, variation in disease prevalence over time, and baseline characteristics and symptom progression using appropriate summary statistics and graphs, with the number of participants with missing data reported separately.

We used two methods to allow for imperfect reference standard bias. Firstly, for individuals with discordant results between either index test and the reference standard, we created an

enhanced composite reference standard using a combination of antibody testing results, additional RT-PCR test results, and linked hospitalisation and mortality records [6]. Secondly, we performed a statistical adjustment to sensitivity and specificity estimates using a Bayesian adjustment approach [13, 14], assuming Beta prior distributions for the sensitivity (prior mean 97%) and specificity (prior mean 99%) of the reference standard derived from performance characteristics of the RT-PCR test in operation during the study period.

To allow for discrepancies between recorded recruitment date and recorded swab dates from some participants, two sensitivity analyses were performed: firstly, a stricter scenario excluding all individuals for whom these dates did not match exactly, and secondly, a less strict scenario in which date discrepancies of up to a week were allowed (as discrepancies may have reflected date recording errors).

Statistical analysis was performed using RStudio 2022.02.3 including the epiR package [15], RStan [16] and WinBUGS14.

## Patient and public involvement

RAPTOR was supported from inception by the CONDOR steering committee public co-chairs for Patient and Public Involvement and Engagement (PPIE) who co-developed the CONDOR platform and its PPIE strategy. Additional PPIE contributors provided pre-funding feedback on the value of the study, reviewed plain language text, and reviewed and co-developed patient information materials. Contributors favourably reviewed the potential burden on patients of involvement in the study.

## Results

### Recruitment

A total of 763 participants consented to recruitment between 29th October 2020 and 12th October 2021 (Fig 1). Two additional potential participants were excluded because they withdrew from consent procedures). Recruitment at the testing centre began on 15th July 2021. At least one POCT result was reported for 738 participants; for the other 25, no reason for the missing POCT result was provided. In the primary analysis of 663 participants, a reference test result could be matched to 245 samples tested with SD Biosensor only, 118 tested with BD Veritor only, and 300 tested with both POCTs.

Fig 2 shows recruitment over the course of the study. Peaks occurred in early 2021 and at the end of summer 2021, the second of which coincided with the start of recruitment at the testing centre. The proportion of the participants with positive RT-PCR results varied throughout the study period, and was highest during periods of elevated recruitment rate.

### Participant characteristics

42% of participants recruited were male, mean age was 41 years, the majority (81%) were white, and 26% were contacts of a household member who had tested positive for SARS-CoV-2 (Table 1). Just over half of the participants had received at least one vaccination dose, with around half of those having received the Oxford-AstraZeneca vaccine. About 21% of those recruited reported a previous SARS-CoV-2 infection. Cough, fatigue, headache and fever were among the most reported baseline symptoms.

### Diagnostic accuracy (primary outcome)

The prevalence of SARS-CoV-2 positive RT-PCR tests among participants included in the primary analysis was 39.2% (260/663, 95% CI 35.5% to 43.0%). The SD Biosensor POCT had a

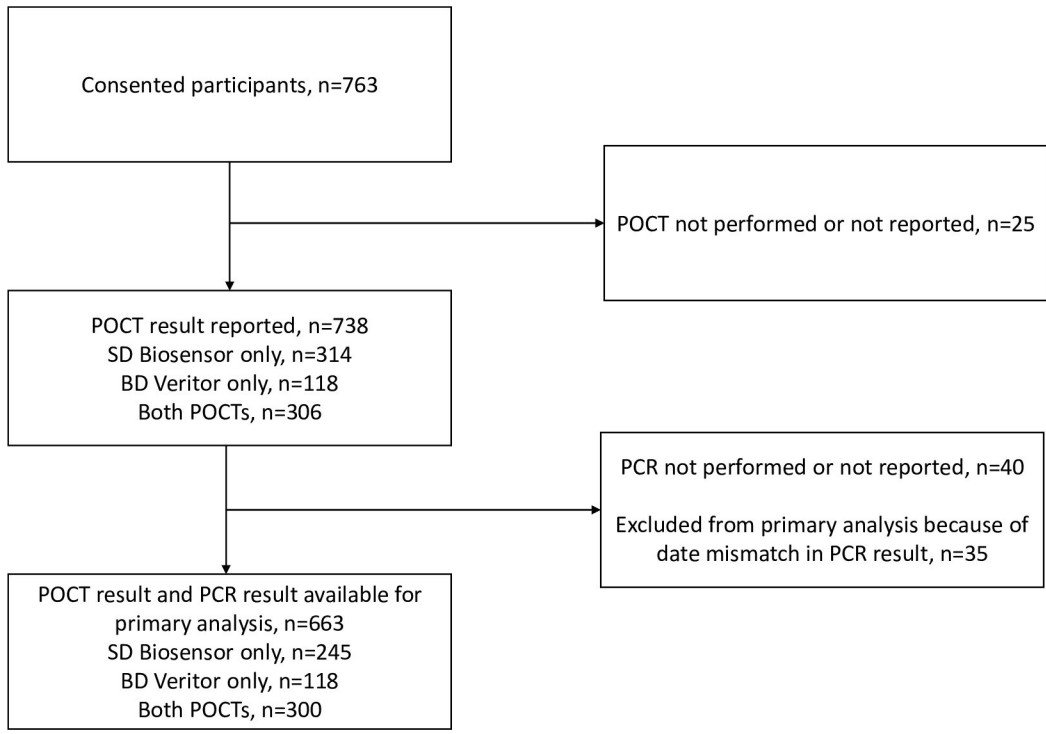

**Fig 1. Recruitment flow chart.**

sensitivity of 84.0% (178/212, 95% CI 78.3% to 88.6%) and specificity of 98.5% (328/333, 96.5% to 99.5%), and the BD Veritor POCT had a sensitivity of 76.5% (127/166, 69.3% to 82.7%) and specificity of 98.8% (249/252, 96.6% to 99.8%) (Table 2 and S1 Table 1 in S1 File). The positive and negative predictive values were 97.3% (178/183, 93.7% to 99.1%) and 90.6% (328/362, 87.1% to 93.4%) respectively for SD Biosensor, and 97.7% (127/130, 93.4% to 99.5%) and 86.5% (249/288, 82.0% to 90.2%) respectively for BD Veritor.

Of the 300 participants who had results for both POCTs and the reference test, 85 had concordant positive results from both POCTs and RT-PCR and 179 had concordant negative results. Patterns of discordance are shown in Table 2. In 17 of the 196 cases when both POCTs gave negative results, the RT-PCR was positive. Among these 17 participants, average time since symptom onset (3.9 days) was similar to that in the full cohort (3.8 days).

Pre-specified subgroup analyses found some variation in test performance according to certain participant characteristics reported (S1 Table 1 in S1 File), with both POCTs having higher sensitivity in males and in participants who reported at least two key symptoms (fever, cough, or change in taste/smell) at baseline. Disease prevalence in the primary analysis cohort was higher in males (47%, 128/272) than in females (34%, 132/391) and much higher in those who reported at least two key symptoms (59%, 151/257).

Among sites that recruited at least 10 participants, sensitivity and specificity estimates were largely similar, although the prevalence of disease varied substantially between sites (S1 Fig 1 in S1 File).

There were two main circulating variants of SARS-CoV-2 during the study period in the UK, VOC Alpha GRY (B.1.1.7+Q.) then VOC Delta GK (B.1.617.2+AY.) (S1 Fig 2 in S1 File). We tracked the performance of both POCTs over time showing that the diagnostic performance of neither test shifted during the transition from one dominant variant to the other (S1

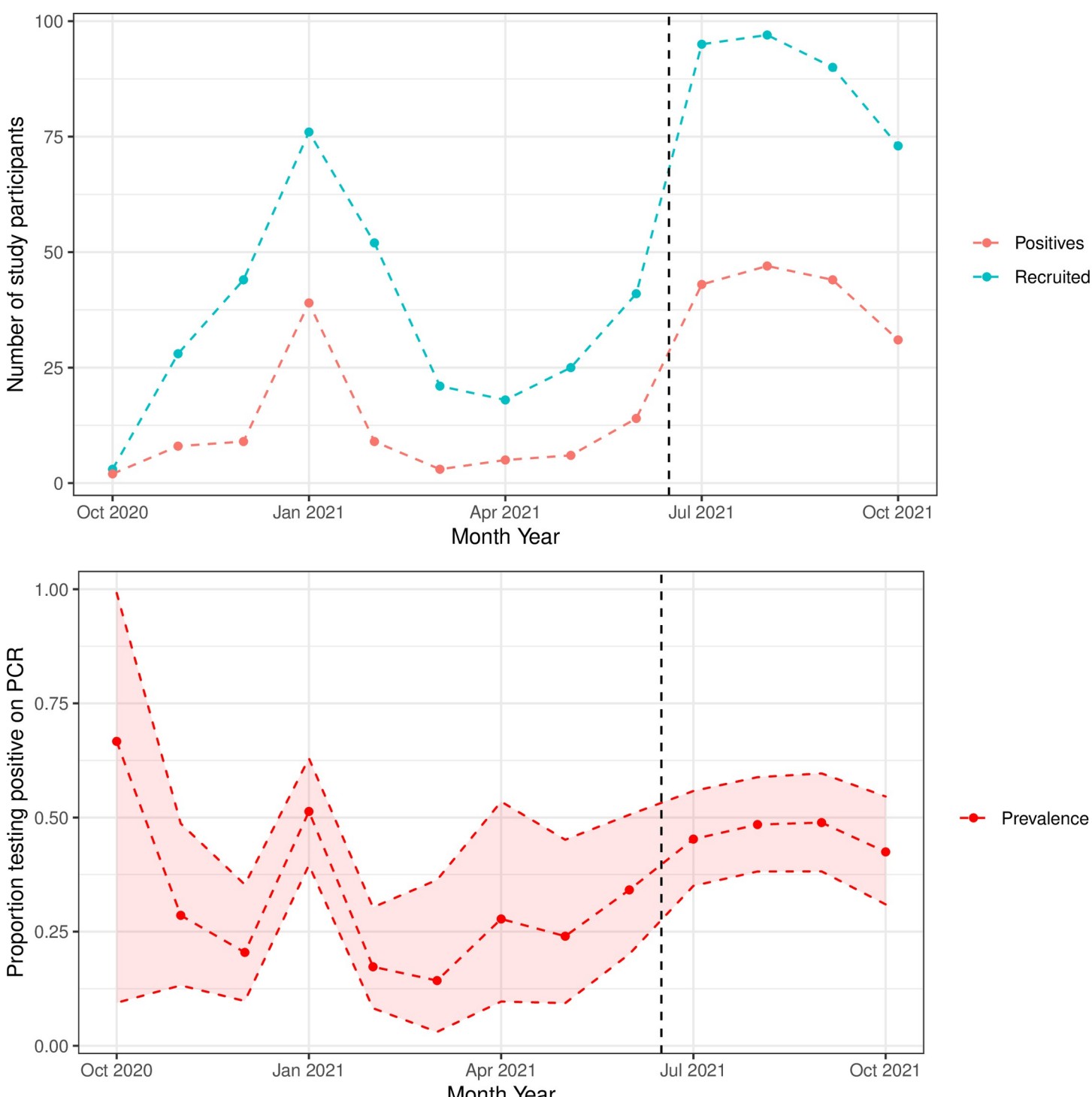

**Fig 2.** Top: monthly recruitment over time (blue), with the number of participants who had a positive reference standard test for SARS-CoV-2 infection (red). Bottom: estimated monthly prevalence of SARS-CoV-2 infection (calculated as the proportion of positive RT-PCR study results) and 95% confidence interval. Data shown refer to the end of the month indicated. The time when the testing centre started recruitment is indicated by the dotted black line.

Figs 3 and 4 in S1 File). Table 3 summarises the diagnostic performance in relation to RT-PCR cycle threshold. Both POCTs show a clear reduction in performance with increasing cycle threshold (reflecting reduced viral load). In an exploratory analysis, the sex differences in

**Table 1. Baseline characteristics (number (%) or mean (standard deviation)).**

| | All participants (n = 763) | Participants included in primary analysis (n = 663) |
|---|---|---|
| Male sex | 317 (42%) | 272 (41%) |
| Age (years) | 41 (19) | 41 (19) |
| • < 16 | 65 (9%) | 52 (8%) |
| • 16–39 | 294 (39%) | 252 (38%) |
| • 40–59 | 263 (34%) | 237 (36%) |
| • 60+ | 141 (18%) | 122 (18%) |
| Ethnicity | | |
| • White | 613 (81%) | 569 (86%) |
| • Asian | 119 (16%) | 69 (10%) |
| • Black | 5 (1%) | 5 (1%) |
| • Mixed-White and Black Caribbean | 4 (1%) | 4 (1%) |
| • Mixed-White and Asian | 5 (1%) | 5 (1%) |
| • Mixed-Other | 8 (1%) | 5 (1%) |
| • Other/not reported | 9 (1%) | 6 (1%) |
| Previous episode of COVID-19 infection | | |
| • Positive RT-PCR test reported | 163 (21%) | 92 (14%) |
| • Number of days since last positive antigen test | 42 (120) | 45 (125) |
| Vaccinated against COVID-19* | 411 (54%) | 389 (59%) |
| • Oxford-AstraZeneca | 210 (51%) | 197 (51%) |
| • Pfizer | 183 (45%) | 175 (45%) |
| • Moderna | 9 (2%) | 8 (2%) |
| • - Other/type unknown | 9 (2%) | 9 (2%) |
| Number of days since first symptom† | 3.8 (2.7) | 3.7 (2.6) |
| Symptoms | | |
| • Any symptom‡ | 717 (94%) | 626 (94%) |
| • Fever | 297 (39%) | 256 (39%) |
| • Cough | 461 (60%) | 410 (62%) |
| • Fatigue | 327 (43%) | 308 (46%) |
| • Shortness of breath | 173 (23%) | 164 (25%) |
| • Sputum | 136 (18%) | 132 (20%) |
| • Loss of smell or change in taste | 167 (22%) | 160 (24%) |
| • Muscle ache | 255 (33%) | 241 (36%) |
| • Chills | 193 (25%) | 184 (28%) |
| • Dizziness | 93 (12%) | 92 (14%) |
| • Headache | 303 (40%) | 288 (43%) |
| • Sore throat | 253 (33%) | 239 (36%) |
| • Hoarseness | 140 (18%) | 139 (21%) |
| • Nausea or vomiting | 114 (15%) | 110 (17%) |
| • Diarrhoea | 62 (8%) | 59 (9%) |
| • Nasal congestion | 244 (32%) | 234 (35%) |
| • Other | 113 (15%) | 110 (17%) |
| Household contact diagnosed with COVID-19 | 201 (26%) | 174 (26%) |

* Vaccinated with at least one dose (booster doses were not recorded)

† Calculated among participants who reported at least one specific symptom within the preceding 14 days

‡ For 6% of participants, no specific symptoms were recorded in the eCRF

diagnostic sensitivity remained when broken down by cycle threshold (S1 Table 2 in S1 File). S1 Fig 5 in S1 File shows the trend in mean cycle threshold across the duration of the study.

There was no clear trend in diagnostic performance in relation to the number of days since first reported symptom among those who commenced participation less than a week after symptom onset, but there was some indication of a decrease in the sensitivity of both index

**Table 2. Summary of results for each POCT compared to the reference test result.**

| | | Reference standard | | | | |
|---|---|---|---|---|---|---|
| | | **Positive** | **Negative** | **Total reported** | **Not reported** | **Date mismatched** |
| SD Biosensor test result | Positive | 178 | 5 | 183 | 7 | 0 |
| | Negative | 34 | 328 | 362 | 33 | 35 |
| | Total | 212 | 333 | 545 | 40 | 35 |
| | | Reference standard | | | | |
| | | Positive | Negative | Total reported | Not reported | Date mismatched |
| BD Veritor test result | Positive | 127 | 3 | 130 | 3 | 0 |
| | Negative | 39 | 249 | 288 | 1 | 2 |
| | Total | 166 | 252 | 418 | 4 | 2 |
| | | Reference standard | | | | |
| | | Positive | Negative | Total reported | Not reported | Date mismatched |
| Both POCT results | SDB positive, BDV positive | 85 | 0 | 85 | 2 | 0 |
| | SDB positive, BDV negative | 12 | 1 | 13 | 1 | 0 |
| | SDB negative, BDV positive | 4 | 2 | 6 | 1 | 0 |
| | SDB negative, BDV negative | 17 | 179 | 196 | 0 | 2 |
| | Total | 118 | 182 | 300 | 4 | 2 |

SDB = SD Biosensor, BDV = BD Veritor.

tests among the small number of participants with positive RT-PCR results and a longer symptom duration (Fig 3).

## Diagnostic accuracy (enhanced reference standard)

A summary of findings using the composite enhanced reference standard among individuals with discordant result between either index test and the RT-PCR reference standard is provided in S1 Table 3 in S1 File. For participants for whom additional information, such as follow-up serology or additional RT-PCR results, was available, this information generally (in 12/14 cases) supported the original reference standard diagnosis.

Results of the statistical adjustment method for imperfect reference standard bias are shown in S1 Fig 6 and S1 Table 4 in S1 File. This adjustment yielded a small increase, of approximately one percentage point, in the estimated sensitivity and specificity of both index tests (SD Biosensor posterior median sensitivity 85.0%, specificity 99.4%, BD Veritor sensitivity 77.4%, specificity 99.4%).

**Table 3. Summary of diagnostic performance in relation to RT-PCR cycle threshold.**

| | | ORF1ab | | | | E gene | | | |
|---|---|---|---|---|---|---|---|---|---|
| | Ct value | $\leq 20$ | 20–25 | 25–30 | $\geq 30$ | $\leq 20$ | 20–25 | 25–30 | $\geq 30$ |
| SD Biosensor | Positive | 58 | 81 | 33 | 6 | 68 | 72 | 33 | 5 |
| | Negative | 0 | 6 | 11 | 15 | 0 | 6 | 12 | 16 |
| | Sensitivity | 1.00 (0.94, 1.00) | 0.93 (0.86, 0.97) | 0.75 (0.60, 0.87) | 0.29 (0.11, 0.52) | 1.00 (0.95, 1.00) | 0.92 (0.84, 0.97) | 0.73 (0.58, 0.85) | 0.24 (0.08, 0.47) |
| BD Veritor | Positive | 59 | 56 | 10 | 1 | 69 | 46 | 10 | 2 |
| | Negative | 1 | 6 | 17 | 14 | 1 | 6 | 18 | 14 |
| | Sensitivity | 0.98 (0.91, 1.00) | 0.90 (0.80, 0.96) | 0.37 (0.19, 0.58) | 0.07 (0.00, 0.32) | 0.99 (0.92, 1.00) | 0.88 (0.77, 0.96) | 0.36 (0.19, 0.56) | 0.12 (0.02, 0.38) |

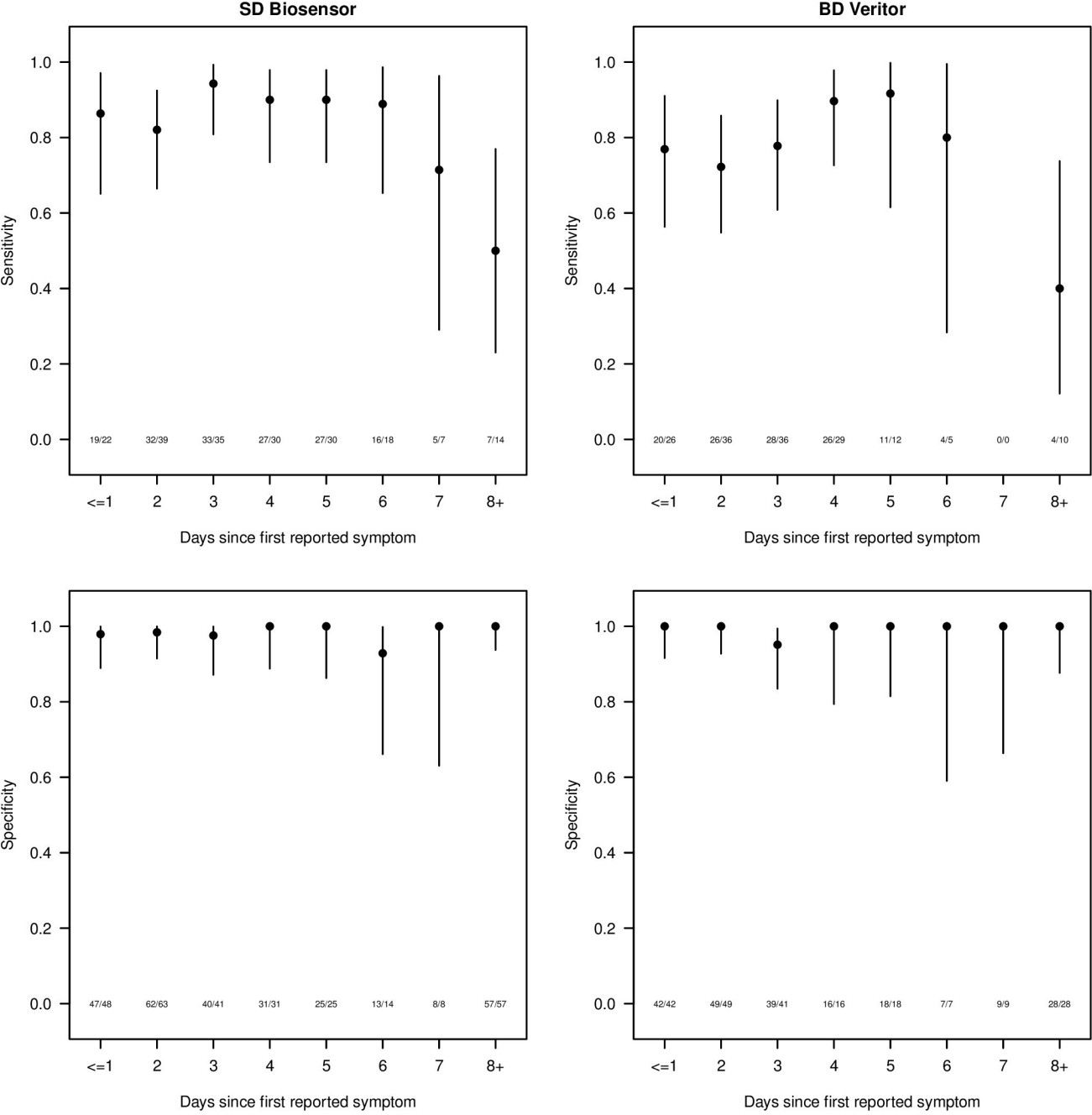

**Fig 3.** Estimated sensitivity (upper two panels) and specificity (lower two panels), with 95% confidence intervals, by number of days since first reported symptom (x-axis), for SD Biosensor (left two panels) and BD Veritor (right two panels). The number of individuals correctly diagnosed by the POCT out of the total are shown towards the bottom of each plot. Participants who did not report specific symptoms and those for whom the timing of symptom onset was unclear are excluded.

## Secondary outcomes

Of the 403 participants who had a negative RT-PCR for SARS-CoV-2, RSV was detected in 12 participants (7 subtype A, 5 subtype B), hMPV in 1 participant and seasonal coronavirus in 8 participants (4 species #NL63, 3 #OC43, 1 #229E). No participants tested positive for influenza

(subtypes A or B). Among the 260 participants who had a positive RT-PCR for SARS-CoV-2, co-infection with RSV was detected in 2 participants (both subtype B), seasonal coronavirus in 2 (both #OC43) and hMPV in 1.

There were no serious adverse events related to study procedures. Three participants were recorded as having been hospitalised within 28 days of a positive COVID-19 test at recruitment with COVID-19 the primary reason for admission, and all were discharged within two weeks. One participant was hospitalised within 28 days of recruitment for an unrelated injury.

### Sensitivity analysis

Sensitivity analyses allowing for different date mismatching scenarios did not show a large impact on estimated diagnostic accuracy measures (S1 Table 5 in S1 File).

## Discussion

The results of this prospective diagnostic accuracy evaluation of two POCTs for the detection of SARS-CoV-2 in symptomatic patients in primary care fall within the wide range of previous studies in other settings [3, 17–22]. A living systematic review found widely varying estimates of the sensitivity for the BD Veritor system (between 41.2% and 96.2% in different studies), and similarly varying estimates for the SD Biosensor system (between 28.6% and 98.3%) [3]. In the primary analysis, we estimate the sensitivities of BD Veritor and SD Biosensor to be 76.5% (95% CI 69.3% to 82.7%) and 84.0% (95% CI 78.3% to 88.6%) respectively. Both devices were found to have specificities close to 99%, which is consistent with most previous studies [23–26].

The minimum target for acceptable performance in the target product profile of the Medicines and Healthcare products Regulatory Agency, is sensitivity of 80% and specificity of 95% [12]. The World Health Organisation target product profile stipulates sensitivity ≥80% and specificity ≥97% [27]. Our results indicate that performance is likely to exceed the specificity threshold, but there remains doubt over performance in relation to sensitivity. Allied to high positive predictive values, this suggests that the most appropriate use of these POCTs may be as rule-in tests, while negative test results do not preclude infection.

Diagnostic performance was strongly associated with RT-PCR cycle threshold. Performance declined at higher cycle thresholds, which are associated with the presence of lower intact sample viral RNA, a proxy for viral load. Test sensitivity declined among individuals whose symptoms began more than one week before recruitment. Correlation has been proposed between higher viral load distributions, LFD positive results and infectiousness of individuals [28], but others have suggested that important numbers of infections may be missed by LFDs due to their limited sensitivity [29]. Without an agreed reference standard for infectiousness, we were unable to assess the value of these tests for identifying infectious individuals [30].

Venekamp et al's community-based study in the Netherlands of tests including SD Biosensor and BD Veritor recruited until June 2021, before the Delta variant became dominant [23]. Our recruitment continued for one year from October 2020, covering the period of the two dominant SARS-CoV-2 variants in the UK circulating during this time (Alpha and Delta). We demonstrated a sustained diagnostic performance for both variants, with sensitivities slightly higher than those reported by Venekamp et al.

Other studies have shown substantial decreases in test sensitivity in asymptomatic individuals, including those recruited as close contacts of cases [25, 31, 32]. Our study demonstrates reduced sensitivity in individuals with fewer core symptoms but does not provide evidence about the performance of the two assays in the asymptomatic population. Based on these findings, patients with a single main symptom (fever, cough or anosmia), could be advised to repeat a negative test if their symptoms persist, or if more symptoms develop.

This study prospectively recruited a large cohort of symptomatic participants attending primary healthcare and two COVID-19 testing centres, and therefore reflects real-world diagnostic accuracy. Understanding performance in primary healthcare is likely to be increasingly important as we cope with waves of endemic infection, and this is one of the few studies to report performance for any POCT in this setting and to our knowledge the only one based in UK primary care.

Recruitment met recommended sample sizes. Further, this study benefitted from contemporaneous swabbing for all tests, and blinding of index test results from those who were performing the reference test (and vice versa), as recommended in diagnostic accuracy studies [33]. A single site performed all reference standard testing to ensure consistency. Our results were adjusted, using two methods, for possible reference standard misclassification and were robust to this adjustment. Paired sampling and use of two index diagnostic tests gave greater scope for direct comparison than previous evaluations.

This study also has some limitations. Because of low recruitment from some sites, a testing centre was added as a recruitment site and so the population tested overall may be less unwell than those who would contact the GP surgery. Prevalence of SARS-CoV-2 infection varied substantially by practice, suggesting there may have been differences in the way in which practices identified participants for recruitment. However, throughout the study recruited patients were required to be symptomatic and diagnostic performance did not change when restricted to participants recruited via the testing centre. This study does not assess diagnostic performance in asymptomatic patients, in whom viral load may be lower and there may be a consequent effect on diagnostic performance. It assesses performance when testing was carried out by clinical staff, rather than via self-swabbing, and performance might decline if not always done according to manufacturers' instructions and performed by a trained operator. Consistent with other studies, we have used RT-PCR cycle threshold as a proxy for viral load and did not apply a calibration and conversion to provide absolute estimates. Fully quantitative assays require a calibrated standard curve, which was not incorporated as an element of this study, as the results were intended to be binary in recognition of how diagnostic decisions are made in the real world.

This study represents 12 months of recruitment, during which time the prevalence of SARS-CoV-2 fluctuated, and results cannot necessarily be extrapolated to future variants should they emerge. For example, some studies have suggested that some assays may have impaired detection for Omicron variants [34].

The number of missing test results was higher than anticipated, and RT-PCR results were unobtainable for 40 samples, most of which were from participants who received the SD Biosensor POCT during the early period of recruitment. The effect of this was explored in sensitivity analyses, which did not show substantial changes in the major results. The principal reason for missing RT-PCR data was because of postal delays during the pandemic period in the early set-up of the study. As such we consider these data to be missing completely at random and do not expect this to bias the results.

In a population with symptoms of COVID-19 presenting to community settings, SD Biosensor and BD Veritor POCTs performed by healthcare professionals are highly specific and so could be used to rule in COVID-19. However the proportion of patients with positive RT-PCR test results who received false negative POCT results was 16.0% for SD Biosensor and 23.5% for BD Veritor, which could result in onward transmission and inappropriate management unless population prevalence of disease is very low. Performance was improved in patients with more symptoms and those with low RT-PCR Ct values. Tests should be interpreted with more caution outside of this clinical phenotype. Though this strategy was not tested, it may be sensible to repeat the POCT in 12 or 24 hours in patients with a clinical

phenotype for COVID-19 who test negative since viral counts may rise over time. This strategy should be studied since identifying true negatives as well as positives is important as waves of this virus continue.

## Supporting information

**S1 File.**
(DOCX)

**S2 File.**
(DOCX)

**S1 Data.**
(DOCX)

## Acknowledgments

We would like to thank the study participants, practice staff at all participating general practice and testing centre sites, and staff of the NIHR Clinical Research Network: Thames Valley & South Midlands (Jithen Benjamin, Joanne Carter, Helen Collins, Mark Dolman, Ross Downes, Kelly Fricker, Kate Hannaby, Heather Kenyon, Kathryn Lucas, Sophie Maslen, Lydia Owen, Cate Wills, Olga Zolle). We also acknowledge the support of Dr Jason Oke and Dr Constanti-nos Koshiaris (University of Oxford) at the planning stage of the study; Helen Bohan and Julian Sherlock (University of Oxford); Kevin Brown, Joanna Ellis, Jamie Lopez Bernal and Tim Brooks (UK Health Security Agency) for assistance interpreting RT-PCR and serology results; Gary Howsam and Victoria Tzortziou-Brown (Royal College of General Practitioners); Dr Matt Wilson, Abi Dhillon and Sian Organ (uMed); patients and practices in the Oxford-Royal College of General Practitioners Research and Surveillance Centre (RSC) who share pseudonymised data to support research and surveillance (UKHSA is the principal sponsor of the RSC); EMIS, TPP, Vision and Wellbeing for assistance with pseudonymised data extraction. We would like to acknowledge GISAID (https://gisaid.org/) and their contributing laboratories for the provision of the SARS-CoV-2 variant epidemiological data displayed in S1 Fig 5 in S1 File.

Here we name the members of the RAPTOR-C19 Study Group[%] and COVID-19 National DiagnOstic Research and Evaluation Platform (CONDOR) Steering Committee[$] as follows: Rachel C. Byford[%1], Alexandra S. Deeks[%1], George Edwards[%1], Jennifer Hirst[%1], Uy Hoang[%1], F. D. Richard Hobbs[%1] (Chief Investigator RAPTOR-C19 Study Group; richard.hobbs@phc.ox.ac.uk), Kirsty Jackson[%1], Heather Kenyon[%*] Joseph J. Lee[%1], Ezra Linley[%2], Mary Logan[%1], Kathryn Lucas[%*], Abigail A. Moore[%1], Lazaro Mwandigha[%1], Meriel Raymond[%2], Praveen Sebastianpillai[%2], Anna E. Seeley[%1], Sharon Tonner[%1], Richard Body[$3] (Co-Chief Investigator CONDOR; richard.body@manchester.ac.uk), Paul Dark[$3], Eloïse Cook[$3], Colette Inkson[$3], Charles Reynard[$3], Gail N. Hayward[%$1] (Co-Chief Investigator CONDOR; gail.hayward@phc.ox.ac.uk), Rafael Perera[%$1], Brian D. Nicholson[%$1], Philip J. Turner[%$1], Peter Buckle[$4], Naoko Jones[$4], Mark Wilcox[$5], Kerrie Davies[$5], Beverley Riley[$5], Adam Gordon[$6], Clare Lendrem[$7], Will Jones[$7], Anna Halstead[$7], A Joy Allen[$7], D Ashley Price[$8], Amanda Winter[$8], Julian Bray-brook[$9], Emily Adams[$10], Valerie Tate[$], Graham Prestwich[$11].

[1]Nuffield Department of Primary Care Health Sciences, University of Oxford, UK
[2]UK Health Security Agency, UK
[3]University of Manchester, UK
[4]Imperial College London, UK
[5]Leeds Teaching Hospitals NHS trust and University of Leeds, UK

[6]University of Nottingham, UK

[7]University of Newcastle upon Tyne, UK

[8] Newcastle upon Tyne Hospitals NHS Foundation Trust, UK

[9]National Measurement Laboratory, UK

[10]Liverpool School of Tropical Medicine, UK

[11]York and Humber AHSN, UK

[*] NIHR Clinical Research Network: Thames Valley & South Midlands, UK

## Author Contributions

**Conceptualization:** Brian D. Nicholson, Philip J. Turner, Gayatri Amirthalingam, Maria Zambon, Richard Body, Rafael Perera, Gail N. Hayward, F.D. Richard Hobbs.

**Data curation:** Thomas R. Fanshawe.

**Formal analysis:** Brian D. Nicholson, Thomas R. Fanshawe.

**Funding acquisition:** Brian D. Nicholson, Philip J. Turner, Gail N. Hayward, F.D. Richard Hobbs.

**Investigation:** Brian D. Nicholson, Philip J. Turner, Thomas R. Fanshawe, Alice J. Williams, Gayatri Amirthalingam, Sharon Tonner, Maria Zambon, Rafael Perera, Gail N. Hayward, F.D. Richard Hobbs.

**Methodology:** Brian D. Nicholson, Philip J. Turner, Thomas R. Fanshawe, Rafael Perera.

**Project administration:** Brian D. Nicholson, Philip J. Turner, Alice J. Williams, Sharon Tonner, Kerrie Davies.

**Resources:** Philip J. Turner, Richard Body, Gail N. Hayward, F.D. Richard Hobbs.

**Supervision:** Brian D. Nicholson, Philip J. Turner, Maria Zambon, Kerrie Davies, Rafael Perera, Simon de Lusignan, Gail N. Hayward, F.D. Richard Hobbs.

**Writing – original draft:** Brian D. Nicholson, Philip J. Turner, Thomas R. Fanshawe.

**Writing – review & editing:** Brian D. Nicholson, Philip J. Turner, Thomas R. Fanshawe, Alice J. Williams, Gayatri Amirthalingam, Sharon Tonner, Maria Zambon, Richard Body, Kerrie Davies, Rafael Perera, Simon de Lusignan, Gail N. Hayward, F.D. Richard Hobbs.

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
