## [Decision Letter · Decision Letter 0]

20 Mar 2023

PONE-D-23-03034Evaluation of the diagnostic accuracy of two point-of-care tests for COVID-19 when used in 2 community settings in the UK primary care COVID diagnostic accuracy platform trial (RAPTOR-C19)PLOS ONE

Dear Dr. Nicholson,

Thank you for submitting your manuscript to PLOS ONE. After careful consideration, we feel that it has merit but does not fully meet PLOS ONE’s publication criteria as it currently stands. Therefore, we invite you to submit a revised version of the manuscript that addresses the points raised during the review process. Your manuscript has been reviewed by three experts in the field of POCT use for the detection of SARS CoV-2  and

two of them suggest that the manuscript should be revised  in order to be acceptable for publication. One of the reviewer is suggesting to reject the manuscript because of poor novelty in a landscape filled with many similar report: I do  not agree with this evaluation, since I believe that the quality of your study is above the level of  many other similar published papers and consequently  I suggest that you undertake a revision procedure  considering the points raised  by the reviewers.One of the most relevant issue is the lack of OMICRON related variants among the viruses incleded: if you feel that this step would not be possible please open a wide discussion on that. I also believe that fact that only symptomatic patients have been included should be widely  and deeply discussed.

Please submit your revised manuscript by May 04 2023 11:59PM.  If you will need more time than this to complete your revisions, please reply to this message or contact the journal office at plosone@plos.org. Please include the following items when submitting your revised manuscript:A rebuttal letter that responds to each point raised by the academic editor and reviewer(s). You should upload this letter as a separate file labeled 'Response to Reviewers'.A marked-up copy of your manuscript that highlights changes made to the original version. You should upload this as a separate file labeled 'Revised Manuscript with Track Changes'.An unmarked version of your revised paper without tracked changes. You should upload this as a separate file labeled 'Manuscript'.

We look forward to receiving your revised manuscript.

Kind regards,

Vittorio Sambri, M.D., Ph.D.

Academic Editor

PLOS ONE

Journal Requirements:

"We have read the journal's policy and the authors of this manuscript have the following competing interests:

The authorship declares funding support for this study from the University of Oxford Medical Sciences Division Benefactors Urgent COVID-19 Fund, the National Institute for Health and Care (NIHR) School of Primary Care Research, and Urgent Public Health funding for the CONDOR Platform from the NIHR and Asthma+Lung UK. The RAPTOR-C19 study team received analysers and assays free of charge from Becton Dickinson for evaluation in this study.

GH declares funding from the National Institute for Health and Care Research (NIHR) paid to the University of Oxford.

KD declares grant funding from Alere Inc and Cepheid Inc paid to her institution for unrelated research. TF declares NIHR support from the NIHR Community Healthcare MIC for diagnostic evaluation research. AJW declares grant funding received by the University of Oxford through a Wellcome Trust Enriching Engagement grant which has supported unrelated patient participation work carried out by the Royal College of General Practitioners Research Surveillance Centre (based at the University of Oxford) for surveillance work. GE declares funding support from the NIHR Community Healthcare MIC recieved by the University of Oxford. AM declares the support of a Wellcome Trust Doctoral Research Fellowship and an NIHR In-practice Fellowship unrelated to this research. PJT declares support from the NIHR Community Healthcare MIC for diagnostic evaluation research. PJT has provided expert support to the Longitude Prize AMR competition administration which is unrelated to this project and for which the University of Oxford received an honorarium. RB declares grant funding for this project from the NIHR and Asthma+Lung UK, with additional funding from the Department of Health and Social Care paid to his host institution. RB declares grants from Siemens Healthineers, Abbott Point-of-Care and Ancon, all paid to his institution for unrelated research. He declares consulting fees received by his institution from Roche, Siemens, Aptamer Group, LumiraDx, Beckman Coulter and Radiometer, with personal fees received from Psyros Diagnostics. RB has received support for attending meetings / travel from Roche and EMCREG International. RB has participated on data safety monitoring boards or advisory boards for the unrelated FORCE Trial, REWIRE Trial, TARGET-CTA, and Magnetocardiography study (MAGNETIC - sponsored by Creavo). RB is the Deputy National Specialty Lead for Trauma & Emergency Care, National Institute for Health and Care Research Clinical Research Network. RB declares receipt of donated reagents for research not detailed in this paper from Roche, LumiraDx, BD, iXensor, Abbott Point-of-Care, Randox, Avacta, Menarini, loan of analysers from Randox and Menarini, and assays run free of charge for research purposes by Chronomics, My110, and Ancon. JJL declares funding from an NIHR Doctoral Research Fellowship which is unrelated to this research. LM declares support from the NIHR Community Healthcare MIC and other NIHR grants to the University of Oxford in support of this work. EL declares unrelated project funding received by the UKHSA Vaccine Evaluation Unit for contract research from GSK, Pfizer and Sanofi. MZ declares her unpaid activities as the Chair of the charitable organisation ISIRV and her membership of the UK SAGE, NERVTAG and JCVI groups."

4. One of the noted authors is a group or consortium [Listed in MS - insufficient characters to complete here.]. In addition to naming the author group, please list the individual authors and affiliations within this group in the acknowledgments section of your manuscript. Please also indicate clearly a lead author for this group along with a contact email address.

Reviewers' comments:

Reviewer's Responses to Questions

**Comments to the Author**

1. Is the manuscript technically sound, and do the data support the conclusions?

Reviewer #1: Yes

Reviewer #2: No

Reviewer #3: Yes

2. Has the statistical analysis been performed appropriately and rigorously? 

Reviewer #1: Yes

Reviewer #2: Yes

Reviewer #3: Yes

3. Have the authors made all data underlying the findings in their manuscript fully available?

Reviewer #1: No

Reviewer #2: Yes

Reviewer #3: Yes

4. Is the manuscript presented in an intelligible fashion and written in standard English?

Reviewer #1: Yes

Reviewer #2: Yes

Reviewer #3: Yes

5. Review Comments to the Author

Reviewer #1: The authors analyse the performance of two POCT tests in a clinical setting on symptomatic individuals with high viral load compare to RT-PCR. The study is planned and carried out thoroughly. The study has a high quality compared to the multitude of studies on this topic but is limited by the restricted sample size, the missing viral load calculation, the lack of Omicron VOC samples as well as asymptomatic individuals.

Major comments:

1. Figure 1: Is there a cause why only in participants tested with SD Biosensor (including participants testes with both tests), PCR results are missing? Please discuss in the limitations.

2. The information contained in the figures in the supplement seems much more interesting to the reader compared to the figures in the manuscript (especially figure 2). Consider creating interesting figures from the supplement data and move the figures from the manuscript to the supplement.

3. Ct value is just a gross measure of viral load. I suggest calculating viral loads from the PCR data as they are comparable to other high quality studies.

4. The study only includes symptomatic persons. This fact should be strengthened during the whole manuscript.

Minor comments:

1. Abstract (only in the Submission form): “(260/663, 95% CI 35.5% to43.0%)” – a space is missing

2. p3, Abstract, Results, why is 95% CI included two times. Consides removing the second or add 95% for all confidence intervals

3. p9, 168-171: Please describe the reference test more in detail. Was it a commercial assay, different assays, …?

4. p10, 190: If you do a sample size calculation, please do it exact. 1500 is quite crude. In my calculation using your assumptions I get the result of 1347

5. p 23, 366-370: The data presented is only performed on symptomatic individuals with high viral loads. Other studies show big differences between symptomatic and asymptomatic individuals. Consider removing the statements on asymptomatic individuals or discuss them together with relevant literature (e.g. https://doi.org/10.1016/j.jinf.2022.12.017
https://doi.org/10.1128/jcm.00991-21 ).

6. Limitations: A limited sensitivity has been reported for the Omicron variants (e.g. https://doi.org/10.1016/j.cmi.2022.08.006
https://doi.org/10.1007/s00430-022-00730-z
https://doi.org/10.1007/s00430-022-00752-7 ). Consider adding this fact to the limitations ot another place in the discussion.

Reviewer #2: THE POOR NOVELTY OF THE PAPER SUGGESTS ITS REJECTION

THERE ARE MANY PAPER ALREADY PUBLISHED ON POCT TESTING FOR COVID-19

THE STUDY DESIGN IS FINE AND THIS IS A WELL-WRITTEN PAPER BUT, IN MY OPINION, IT DOES NOD ADD VALUABLE INFORMATION TO CURRENT KNOWLEDGE

Reviewer #3: General comments:

The manuscript thoroughly describes the performance of two different antigen tests (one with visual reading of results and one with machine reading of results) in an outpatient setting. Although the paper could be considered post festum (the current SARS-CoV-2 variants were not present during the testing period), the manuscript highlights, documents and adequately discuss the inferior diagnostic accuracy of antigen testing compared to Gold Standard qPCR testing. The manuscript is well written with huge amounts of clinical data and adequate statistics.

Specific comments:

Line 68: I do not think LFD-Ag are commonplace for community testing at the present time?

Methods: A description of the two assays is warranted – e.g. the difference between manual and machine reading of results.

6. PLOS authors have the option to publish the peer review history of their article (what does this mean?). If published, this will include your full peer review and any attached files.

Reviewer #1: No

Reviewer #2: **Yes: **MARIO PLEBANI

Reviewer #3: No

---

## [Author Response · Author response to Decision Letter 0]

5 Jun 2023

05th June 2023

Dear Professor Sambri,

We hope that this letter finds you well. We were very grateful to you for your editorial comments and guidance and to the three reviewers for their considered and constructive reviews of our manuscript.

We are delighted to respond to your comments and those of the reviewers in the following document and through submission of a revised manuscript and associated material. We trust that our revisions will be acceptable to you and the reviewers.

Yours sincerely,

Brian Nicholson, Tom Fanshawe & Phil Turner, on behalf of the authorship

Response to the editor and reviewers

PONE-D-23-03034

Evaluation of the diagnostic accuracy of two point-of-care tests for COVID-19 when used in 2 community settings in the UK primary care COVID diagnostic accuracy platform trial (RAPTOR-C19)

One of the most relevant issue is the lack of OMICRON related variants among the viruses incleded: if you feel that this step would not be possible please open a wide discussion on that. I also believe that fact that only symptomatic patients have been included should be widely and deeply discussed.

Response: We are grateful to Professor Sambri for these observations and have amended the manuscript title to make clear from the outset that the evaluation focused on symptomatic participants. The symptomatic focus of the study is also referenced in the abstract and discussion. This is further discussed in response to Reviewer #1 point 4 below.

We have made specific reference to a publication in the Discussion which describes variable detection of Omicron by rapid antigen tests (Osterman A, Badell I, Dächert C, Schneider N, Kaufmann A-Y, Öztan GN, et al. Variable detection of Omicron-BA.1 and -BA.2 by SARS-CoV-2 rapid antigen tests. Medical Microbiology and Immunology. 2023;212(1):13-23. doi: 10.1007/s00430-022-00752-7.), at the point where we describe the possibility that it may not be possible to extrapolate our results to Omicron or future variants of SARS-CoV-2.

Journal Requirements:

Response: We have checked our resubmission against PLOS ONE’s style guide and are confident that it conforms. We have applied the PLOS ONE endnote style to in-text citations and the reference list, with in-text references now appearing within square parentheses throughout the manuscript.

Response: We have moved this information to the ethics paragraph and provided more details.

"We have read the journal's policy and the authors of this manuscript have the following competing interests:

The authorship declares funding support for this study from the University of Oxford Medical Sciences Division Benefactors Urgent COVID-19 Fund, the National Institute for Health and Care (NIHR) School of Primary Care Research, and Urgent Public Health funding for the CONDOR Platform from the NIHR and Asthma+Lung UK. The RAPTOR-C19 study team received analysers and assays free of charge from Becton Dickinson for evaluation in this study.

GH declares funding from the National Institute for Health and Care Research (NIHR) paid to the University of Oxford.

KD declares grant funding from Alere Inc and Cepheid Inc paid to her institution for unrelated research. TF declares NIHR support from the NIHR Community Healthcare MIC for diagnostic evaluation research. AJW declares grant funding received by the University of Oxford through a Wellcome Trust Enriching Engagement grant which has supported unrelated patient participation work carried out by the Royal College of General Practitioners Research Surveillance Centre (based at the University of Oxford) for surveillance work. GE declares funding support from the NIHR Community Healthcare MIC recieved by the University of Oxford. AM declares the support of a Wellcome Trust Doctoral Research Fellowship and an NIHR In-practice Fellowship unrelated to this research. PJT declares support from the NIHR Community Healthcare MIC for diagnostic evaluation research. PJT has provided expert support to the Longitude Prize AMR competition administration which is unrelated to this project and for which the University of Oxford received an honorarium. RB declares grant funding for this project from the NIHR and Asthma+Lung UK, with additional funding from the Department of Health and Social Care paid to his host institution. RB declares grants from Siemens Healthineers, Abbott Point-of-Care and Ancon, all paid to his institution for unrelated research. He declares consulting fees received by his institution from Roche, Siemens, Aptamer Group, LumiraDx, Beckman Coulter and Radiometer, with personal fees received from Psyros Diagnostics. RB has received support for attending meetings / travel from Roche and EMCREG International. RB has participated on data safety monitoring boards or advisory boards for the unrelated FORCE Trial, REWIRE Trial, TARGET-CTA, and Magnetocardiography study (MAGNETIC - sponsored by Creavo). RB is the Deputy National Specialty Lead for Trauma & Emergency Care, National Institute for Health and Care Research Clinical Research Network. RB declares receipt of donated reagents for research not detailed in this paper from Roche, LumiraDx, BD, iXensor, Abbott Point-of-Care, Randox, Avacta, Menarini, loan of analysers from Randox and Menarini, and assays run free of charge for research purposes by Chronomics, My110, and Ancon. JJL declares funding from an NIHR Doctoral Research Fellowship which is unrelated to this research. LM declares support from the NIHR Community Healthcare MIC and other NIHR grants to the University of Oxford in support of this work. EL declares unrelated project funding received by the UKHSA Vaccine Evaluation Unit for contract research from GSK, Pfizer and Sanofi. MZ declares her unpaid activities as the Chair of the charitable organisation ISIRV and her membership of the UK SAGE, NERVTAG and JCVI groups. This does not alter our adherence to PLOS ONE policies on sharing data and materials."

Response: Please see our amended ‘Competing Interests Statement’ above; we have also included this text in the covering letter. We are not aware of any impediments associated with the declaration which would prevent us from complying with PLOS ONE policies on sharing data and materials.

4. One of the noted authors is a group or consortium [Listed in MS - insufficient characters to complete here.]. In addition to naming the author group, please list the individual authors and affiliations within this group in the acknowledgments section of your manuscript. Please also indicate clearly a lead author for this group along with a contact email address.

Response: The original submission listed group/consortium membership and institutional affiliation in the acknowledgements section. We have amended this section as we had inadvertently omitted Dr Sharon Tonner and Professor F.D. Richard Hobbs from the RAPTOR-C19 Study Group and we failed to assign an affiliation to Mary Logan. We have also marked the leads of these groups as ‘Chief Investigator’ and ‘Co-chief Investigator’ for the RAPTOR-C19 Study Group and CONDOR respectively. We have provided email addresses for ‘Chief Investigator’ and ‘Co-chief Investigators’.

Response: The datasets used and analysed during the current study contain potentially sensitive and identifiable patient information under the definitions of UK data protection legislation. Requests for de-identified participant level data collected during this study should be made to the Nuffield Department of Primary Care hosted Datasets Independent Scientific Committee (PrimDISC): primdisc@phc.ox.ac.uk . Data will be released following review and approval by PrimDISC of a protocol, statistical analysis plan and the signing of a suitable data sharing agreement.

Review Comments to the Author

Reviewer #1: The authors analyse the performance of two POCT tests in a clinical setting on symptomatic individuals with high viral load compare to RT-PCR. The study is planned and carried out thoroughly. The study has a high quality compared to the multitude of studies on this topic but is limited by the restricted sample size, the missing viral load calculation, the lack of Omicron VOC samples as well as asymptomatic individuals.

Major comments:

1. Figure 1: Is there a cause why only in participants tested with SD Biosensor (including participants testes with both tests), PCR results are missing? Please discuss in the limitations.

Response: Thank you for this comment. The main reason that more RT-PCR results were missing among participants tested with the SD Biosensor POCT than the BD Veritor POCT was because of postal issues in the delivery of samples for RT-PCR testing during the pandemic period in the early part of recruitment (SD Biosensor was the first POCT to come on board the study). Some samples were inadvertently delivered to the wrong location by the postal service and others were lost in transit. As this is the main reason for this missing data we do not expect this to be related to the RT-PCR result and so would consider these missing results not to bias estimates of diagnostic accuracy. We have explained this in the relevant paragraph in the limitations part of the Discussion.

2. The information contained in the figures in the supplement seems much more interesting to the reader compared to the figures in the manuscript (especially figure 2). Consider creating interesting figures from the supplement data and move the figures from the manuscript to the supplement.

Response: We feel it is somewhat subjective as to which figures readers would find more interesting, but we agree that the previous S1 Fig 6 (diagnostic accuracy in relation to time since first reported symptom) would have been better in the main manuscript (now Fig 3).

3. Ct value is just a gross measure of viral load. I suggest calculating viral loads from the PCR data as they are comparable to other high quality studies.

Response: We thank the reviewer for making this point. We prefer that our data remain in the format of Ct values, which are the common lexicon for these kinds of studies. Fully quantitative assays require a calibrated standard curve, which was not included in this study, as the results were intended to be binary. The comparator of positive/negative is indicative of how diagnostic decisions are made in the real world, where absolute viral load and even Ct values are not used by clinicians. Our study is consistent with many published reports. Most other studies that have evaluated POCTs in other settings, and/or systematic reviews, report by Ct value and not directly by viral load measures (such as the living systematic review, Brümmer et al., 2021. Accuracy of novel antigen rapid diagnostics for SARS-CoV-2: PLoS Med, 18(8), p.e1003735). We have added a note in the limitation paragraph of the Discussion to highlight these points. 

4. The study only includes symptomatic persons. This fact should be strengthened during the whole manuscript.

Response: Participants with symptoms consistent with SARS-CoV-2 were the target patient group for this study. We had previously noted this in several places of the manuscript (including the Abstract Methods section, final sentence of Introduction, Recruitment and participant eligibility section of Methods, several places in Discussion including a comparison of our results with other studies performed in asymptomatic individuals, and in the limitations paragraph of the Discussion). However, we agree that it is important to emphasise further so have now made this additionally clear in the title of the manuscript, the Conclusions section of the Abstract and the first sentence of the Discussion.

Minor comments:

1. Abstract (only in the Submission form): “(260/663, 95% CI 35.5% to43.0%)” – a space is missing

Response: We have corrected this in the Submission form.

2. p3, Abstract, Results, why is 95% CI included two times. Consides removing the second or add 95% for all confidence intervals

Response: We have corrected this.

3. p9, 168-171: Please describe the reference test more in detail. Was it a commercial assay, different assays, …?

Response: We have expanded the ‘Reference standard’ sub-section of the ‘Methods’ to provide more detail on the RT-PCR assay used for reference testing in the study.

4. p10, 190: If you do a sample size calculation, please do it exact. 1500 is quite crude. In my calculation using your assumptions I get the result of 1347

Response: The primary target sample size was 150 positive cases (in line with the UK Medicines and Healthcare products Regulatory Agency (MHRA) Target Product Profile) and the 1500 figure was only ever intended as an approximate total target sample size based on an assumed prevalence of 10%. For the reasons outlined, this total in any case needed to be adjusted as a result of the fluctuating and unpredictable prevalence of SARS-CoV-2 infection over the period of the study. Full details of the sample size calculation are available in the (cited) published protocol and the Statistical Analysis Plan that was provided as a supplementary file.

5. p 23, 366-370: The data presented is only performed on symptomatic individuals with high viral loads. Other studies show big differences between symptomatic and asymptomatic individuals. Consider removing the statements on asymptomatic individuals or discuss them together with relevant literature (e.g. https://doi.org/10.1016/j.jinf.2022.12.017
https://doi.org/10.1128/jcm.00991-21 ).

Response: We agree that this study does not demonstrate performance in asymptomatic individuals, in whom performance may differ from symptomatic individuals, and had already stated in the limitations section of the Discussion: “This study does not assess diagnostic performance in asymptomatic patients, in whom viral load may be lower and there may be a consequent effect on diagnostic performance.”

Also in the Discussion we state “Other studies have shown substantial decreases in test sensitivity in asymptomatic individuals, including those recruited as close contacts of cases” and cite three papers to support this. To make this additionally clear we now also state “Our study demonstrates reduced sensitivity in individuals with fewer core symptoms but does not provide evidence about the performance of the two assays in the asymptomatic population.”

However we do not feel the two papers indicated by the reviewer are directly relevant as they report results of other assays which were not evaluated in our study.

6. Limitations: A limited sensitivity has been reported for the Omicron variants (e.g. https://doi.org/10.1016/j.cmi.2022.08.006
https://doi.org/10.1007/s00430-022-00730-z
https://doi.org/10.1007/s00430-022-00752-7 ). Consider adding this fact to the limitations ot another place in the discussion.

Response: We are grateful to the reviewer for this suggestion and have added this statement and references about possible impaired performance for Omicron variants to the limitations part of the Discussion that mentions future SARS-CoV-2 variants.

Reviewer #2: THE POOR NOVELTY OF THE PAPER SUGGESTS ITS REJECTION

THERE ARE MANY PAPER ALREADY PUBLISHED ON POCT TESTING FOR COVID-19

THE STUDY DESIGN IS FINE AND THIS IS A WELL-WRITTEN PAPER BUT, IN MY OPINION, IT DOES NOD ADD VALUABLE INFORMATION TO CURRENT KNOWLEDGE

Response: Thank you for your review. We respectfully disagree as there are no other published studies of this size, and with the methodological strengths of this study, conducted in community settings.

Reviewer #3: General comments:

The manuscript thoroughly describes the performance of two different antigen tests (one with visual reading of results and one with machine reading of results) in an outpatient setting. Although the paper could be considered post festum (the current SARS-CoV-2 variants were not present during the testing period), the manuscript highlights, documents and adequately discuss the inferior diagnostic accuracy of antigen testing compared to Gold Standard qPCR testing. The manuscript is well written with huge amounts of clinical data and adequate statistics.

Response: Thank you for your considered review of the manuscript.

Specific comments:

Line 68: I do not think LFD-Ag are commonplace for community testing at the present time?

Response: We have rewritten the opening sentences of the manuscript to reflect changes in community testing practices and have removed original reference [1] accordingly.

Methods: A description of the two assays is warranted – e.g. the difference between manual and machine reading of results.

Response: We have added additional descriptive detail with respect to the two index tests under the ‘Index tests’ heading, highlighting the key differences in test result interpretation i.e. manual/user vs machine.

---

## [Editor Report · Decision Letter 1]

2 Jul 2023

Evaluation of the diagnostic accuracy of two point-of-care tests for COVID-19 when used in symptomatic patients in community settings in the UK primary care COVID diagnostic accuracy platform trial (RAPTOR-C19)

PONE-D-23-03034R1

Dear Dr. Nicholson,

We’re pleased to inform you that your revised manuscript has been now judged scientifically suitable for publication and will be formally accepted for publication once it meets all outstanding technical requirements. Thank you for the efforts made to revise your work accordingly to the suggestions made by the reviewers.

Kind regards,

Vittorio Sambri, M.D., Ph.D.

Academic Editor

PLOS ONE

---

## [Editor Report · Acceptance letter]

13 Jul 2023

PONE-D-23-03034R1 

Evaluation of the diagnostic accuracy of two point-of-care tests for COVID-19 when used in symptomatic patients in community settings in the UK primary care COVID diagnostic accuracy platform trial (RAPTOR-C19) 

Dear Dr. Nicholson:

I'm pleased to inform you that your manuscript has been deemed suitable for publication in PLOS ONE. Congratulations! Your manuscript is now with our production department. 

Kind regards, 

on behalf of

Professor Vittorio Sambri 

Academic Editor

PLOS ONE